



# Driving factors for the activity coefficient of atmospheric ammonium nitrate: discrepancies among thermodynamic models and impact on nitrate pollutions

Ruilin Wan, Guangjie Zheng*, Yuyang Li, Xiaolin Duan, Jingkun Jiang, and Kebin He

State Key Laboratory of Regional Environment and Sustainability, School of Environment, Tsinghua University, Beijing 100084, China

*Correspondence to*: G. Zheng (zgj123@mail.tsinghua.edu.cn)

**Abstract.** Semi-volatile $NH_4NO_3$ is a major component of atmospheric aerosols, and its environmental and climate effects are largely regulated by the gas-particle partitioning. The activity coefficient of $NH_4NO_3$, $\gamma_{AN}$, is one key parameter controlling

the gas-particle partitioning, yet its dependence on meteorological condition and chemical profile remains uncertain. Here we investigated into this issue with comprehensive simulations and ambient observations, based on results of three widely-used thermodynamic models, i.e. ISORROPIA, E-AIM, and AIOMFAC. Across all models, $\gamma_{AN}^2$ ranges between $10^{-2}$ and $10^{-1}$, with AIOMFAC results ~ 33% lower than E-AIM and ISORROPIA. Correspondingly, AIOMFAC estimate higher particle phase nitrate $f_{p,NO_3^-}$ values. The $\gamma_{AN}^2$ correlates positively with relative humidity (RH) and temperature, while RH generally

contributes larger variations under typical scenarios. The effect of chemical composition on $\gamma_{AN}^2$ is more complex and is strongly modulated by RH. Furthermore, $\gamma_{AN}^2$ responds more strongly to changes of particle chemical profile in E-AIM, whereas in ISORROPIA and AIOMFAC $\gamma_{AN}^2$ is more sensitive to meteorological variations. As E-AIM is typically considered as the benchmark thermodynamic model, these results suggest the potential under-representation of chemical profiles in predicting $\gamma_{AN}^2$ for ISORROPIA and AIOMFAC. The corresponding influence on 3-D chemical-transport model predictions

of $NH_4NO_3$ are encouraged in future studies.



**Graphical abstract:**

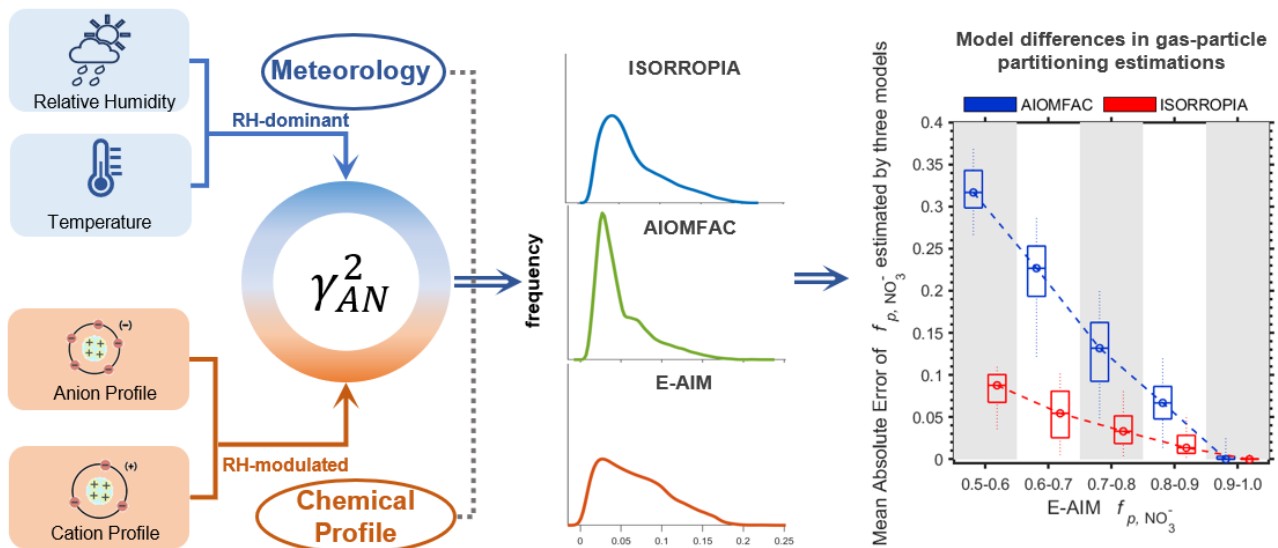



## 1 Introduction

Nitrate is a key component of atmospheric aerosols, exerting substantial influence on haze formation and climate(Li et al.,
2019, 2023; Wang et al., 2024; Xu et al., 2019). As nitrate is semi-volatile, the gas–particle partitioning process plays a critical
role in regulating the particulate nitrate concentrations(Qi et al., 2023; Zhai et al., 2021), new particle formation and growth(Li
et al., 2024; Wang et al., 2022), global nitrogen deposition rates(Arangio et al., 2022; Nenes et al., 2021; Pan et al., 2024), and
the atmospheric photochemical oxidative capacity(Cao et al., 2023; Shi et al., 2021; Ye et al., 2017; Zhang et al., 2025). Nitrate

gas–particle partitioning is governed by the interplay of gas–liquid equilibrium, charge balance, acid dissociation equilibrium,
and the non-ideality of aerosol solution(Guo et al., 2015, 2017b; Nenes et al., 2020, 2021; Pye et al., 2020b). Due to its
complexity, the mechanisms and influencing factors of nitrate gas-particle partitioning are still not fully understood, as
indicated by the discrepancy between observations and model simulations(Guo et al., 2015, 2017a), and among different
thermodynamic models. Inaccurate estimation of nitrate gas-particle partitioning is one major source of simulation uncertainty

for nitrate concentration and its environmental and climate effects (Mezuman et al., 2016; Nault et al., 2021; Norman et al.,
2025).

Among the potential influencing factors, non-ideality is the one with the largest uncertainty. Non-ideality refers to the degree
to which the thermodynamic properties of a solution deviate from the behavior of an ideal solution, which is typically
quantitatively described by the activity coefficient $\gamma$. Conditions such as high ionic strength and increased solution complexity

(e.g., coexistence of multiple organic and inorganic species) can drive $\gamma$ away from unity(Atkins et al., 2023). Deliquescent
atmospheric aerosols are highly concentrated solutions with strong non-ideality(Clegg et al., 1998a, c). However, in-situ
measurement of $\gamma$ for ambient aerosols is challenging due to the extremely high ionic strengths, the complex and varied aerosol
compositions, the low concentrations and therefore high measurement uncertainties for relevant species, etc. (Li et al., 2022;
Nenes et al., 1998; Pitzer, 1987). Consequently, the non-ideality for aerosols is typically estimated by state-of-the-art

thermodynamic models.

Three thermodynamic models are widely adopted to estimate non-ideality in aerosols, i.e. the ISORROPIA (Fountoukis and
Nenes, 2007; Nenes et al., 1998), the Extended Aerosol Inorganics Model (E-AIM)(Friese and Ebel, 2010; Wexler and Clegg,
2002) and Aerosol Inorganic-Organic Mixtures Functional groups Activity Coefficients (AIOMFAC)(Zuend et al., 2008,
2011). These models typically incorporate factors such as ionic strength, electrostatic interactions, and organic–inorganic

coupling to enhance the accuracy of simulations, but the detailed assumptions differed. The ISORROPIA employs an extended
Debye-Hückel form ("Bromley's formula") with empirical ion-pair parameters for ionic strength up to ~6 mol kg$^{-1}$.(Bromley,
1973; Nenes et al., 1998) The E-AIM calculated $\gamma$ for individual ions based on the Pitzer–Simonson–Clegg formula, which
accounted for long-range electrostatic interactions via Debye-Hückel effect and short-range binary/ternary ion–ion interactions
through a Margules expansion(Clegg et al., 1992; Pitzer and Simonson, 1986), with parameters from empirical data(Carslaw

et al., 1995; Clegg et al., 1998b; Friese and Ebel, 2010). AIOMFAC combines a Pitzer-like electrolyte model with a modified
UNIFAC approach, representing long-, middle-, and short-range organic–inorganic interactions(Zuend et al., 2010; Zünd,





2007). E-AIM and ISORROPIA include gas–liquid equilibrium modules(Clegg et al., 2008; Clegg and Brimblecombe, 1990; Wexler and Clegg, 2002) and use the Zdanovskii-Stokes-Robinson method for aerosol water content (AWC)[21,22], whereas AIOMFAC doesn't perform gas–particle phase-equilibrium solving and predicts water activity directly as RH (Seinfeld and Pandis, S. N., 2016; Zuend et al., 2008). Generally, E-AIM is considered as the most accurate "benchmark" model, and ISORROPIA is optimized for computing speed and is widely adopted in chemical transport models, while AIOMFAC offers the strongest capability for inorganic–organic interaction predictions(Hull et al., 2025; Li et al., 2022; Seinfeld and Pandis, S. N., 2016). In atmospheric aerosols, the $NO_3^-$ is usually neutralized by $NH_4^+$ and exist in the form of $NH_4NO_3$(Nowak et al., 2010; Pathak et al., 2009; Seinfeld and Pandis, S. N., 2016). Our previous studies have revealed that the mean activity coefficient of ammonium nitrate, $\gamma_{AN} = \sqrt{\gamma_{NH_4^+}\gamma_{NO_3^-}}$, is a key parameter influencing the gas-particle partitioning of nitrate (see SI Text S1)(Zheng et al., 2022). However, previous studies on thermodynamic model comparison and performance evaluations on non-ideality characterizations focused primarily on acidity (i.e., the activity coefficient of $H^+$) (Liu et al., 2017; Peng et al., 2019; Song et al., 2018; Yao et al., 2006; Zheng et al., 2022), while their performance on $\gamma_{AN}^2$ estimation is less investigated. To bridge this gap, we examined into activity coefficient of atmospheric ammonium nitrate based on both simulated cases and worldwide ambient data. The dependences of $\gamma_{AN}^2$ on different meteorological conditions and chemical profiles are compared among three thermodynamic models of ISORROPIA, E-AIM and AIOMFAC. The $\gamma_{AN}^2$ variability across different regions are further assessed through tests of worldwide observation data. The implications on global nitrate estimations and atmospheric chemistry are also discussed.

## 2 Data and Method

### 2.1 Running different thermodynamic models

Three thermodynamic models were utilized to simulate the non-ideality in aerosols, i.e. the ISORROPIA (v2.3) (Fountoukis and Nenes, 2007; Nenes et al., 1998), E-AIM (version IV)(Friese and Ebel, 2010; Wexler and Clegg, 2002), and AIOMFAC(Zuend et al., 2008, 2011). However, to enable the direct comparison of results among these three models, a set of pre- and post-processing are required to harmonize their inputs and outputs. The overall flow chart is shown in Fig. 1.

Inputs of ISORROPIA and E-AIM are similar, which are the total (gas + particle) concentrations, relative humidity (RH) and temperature. Note here we run both models in forward and metastable modes. However, as E-AIM is unable to explicitly treat all crustal species (e.g., $Ca^{2+}$, $Mg^{2+}$, $K^+$), these species are converted to charge-equivalent $Na^+$ in model comparison studies (Parworth et al., 2017; Peng et al., 2019). In comparison, inputs of AIOMFAC require condensed-phase concentrations together with AWC, which can be acquired from the outputs of either ISORROPIA or E-AIM. Our tests show that the estimated AWC agreed well between these two models, while E-AIM generally provides a more balanced ionic output, particularly in $Na^+$-$NH_3$-$H_2SO_4$-$HNO_3$-$H_2O$ scenario (see SI Text S2 and Fig. S1). Therefore, the predicted condensed-phase concentrations and AWC from E-AIM are used as the inputs for AIOMFAC in subsequent calculation.



The gas-particle partitioning of HNO$_3$ can be represented by $f_{p,\text{NO}_3^-}$, namely the molar fraction of particle-phase NO$_3^-$ in total nitric acid HNO$_{3,\text{tot}}$ as:

$$f_{p,\text{NO}_3^-} = \frac{[\text{NO}_3^-(p)]}{[\text{total HNO}_3]} = \frac{[\text{NO}_3^-(p)]}{[\text{NO}_3^-(p)]+[\text{HNO}_3(g)]} \tag{1}$$

where [X] hereinafter represents the molar concentration of species $X$(µmol/m³).

The $f_{p,\text{NO}_3^-}$ is directly estimated in ISORROPIA and E-AIM, while AIOMFAC does not directly provide gas-particle partitioning results. Therefore, for AIOMFAC the $f_{p,\text{NO}_3^-}$ is calculated in a similar approach to that described by Pye et al.(Pye et al., 2018) as:

$$f_{p,\text{NO}_3^-} = 1 - \frac{p}{RT} \cdot \frac{m_\text{H}+\gamma_\text{H}+m_{\text{NO}_3^-}\gamma_{\text{NO}_3^-}}{K_{\text{HNO}_3}n_{\text{NO}_3}^{\text{total}}} \tag{2}$$

where $m_i$ is molality of ion $i$ (mol·kg⁻¹ water) and $\gamma_i$ is molality-based activity coefficient of ion $i$. The $p$ is ambient pressure in Pa, normally taken as 101325 Pa. The $T$ is absolute temperature in Kelvin, $R$ refers to universal gas constant with a value of 8.314 J·mol⁻¹·K⁻¹, $K_{\text{HNO}_3}$ is the temperature-dependent equilibrium constant of specie HNO$_3$ (see Table S1), and $n_{\text{NO}_3}^{\text{total}}$ is the total (gas and particle phase) concentration of HNO$_3$ in mol m⁻³.

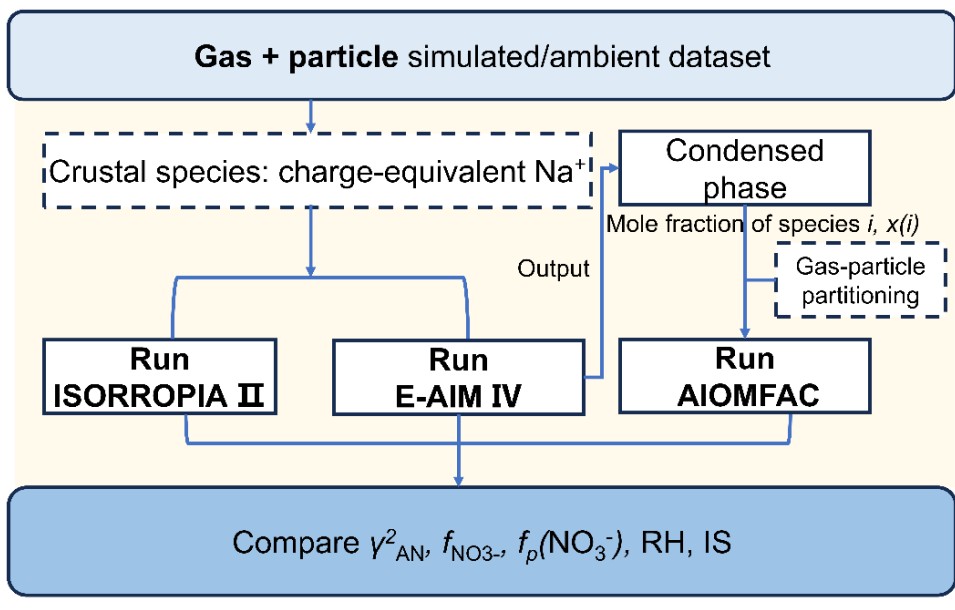

**Figure 1.** **Flow chart of comparison experiments among the three thermodynamic models of ISORROPIA, E-AIM and AIOMFAC.**

### 2.2 Scenario settings for thermodynamic model evaluations

Here we investigated into the potential influencing factors of $\gamma_{AN}^2$ for two aerosol systems, i.e. the NH$_3$-H$_2$SO$_4$-HNO$_3$-H$_2$O system and the Na$^+$-NH$_3$-H$_2$SO$_4$-HNO$_3$-H$_2$O system. The former system is frequently adopted in chamber experiments and




simplified theoretical calculations, as they represented major aerosol compositions of $(NH_4)_2SO_4$ and $NH_4NO_3$ (Seinfeld and Pandis, S. N., 2016; Weber et al., 2016). The latter system is designed to represent the ambient aerosol systems, as global inorganic aerosol components are dominated by ammonia sulfate and ammonia nitrate with crustal species existent(Liu et al., 2025).

Several representative scenarios were set up to examine the effect of meteorological condition, chemical profile and their relative importance on $\gamma_{AN}^2$. The Scenario SNA is designed for the $NH_3$-$H_2SO_4$-$HNO_3$-$H_2O$ system, while the others are based on the $Na^+$-$NH_3$-$H_2SO_4$-$HNO_3$-$H_2O$ system. The anion profile is represented by $f_{NO3^-}$, defined as the molar ratio of $NO_3^-$ to total anions (Eq. 3a). The cation profile is represented by $f_{NVC}$, defined as the molar ratio of $Na^+$ to total cations (Eq. 3b) as:

$$f_{NO_3^-}\left(\frac{\mu eq}{\mu eq}\right) = \frac{[NO_3^-(p)]}{[Anions(p)]} = \frac{[NO_3^-(p)]}{[NO_3^-(p)]+2[SO_4^{2-}(p)]} \tag{3a}$$

$$f_{NVCs}\left(\frac{\mu eq}{\mu eq}\right) = \frac{[NVCs(p)]}{[Cations(p)]} = \frac{[Na^+(p)]}{[Na^+(p)]+[NH_4^+(p)]} \tag{3b}$$

The detailed scenario settings are listed below.

**Scenario SNA:** This scenario examines $\gamma_{AN}^2$ in the absence of $Na^+$. For this system, the particle phase contains only $(NH_4)_2SO_4$ and $NH_4NO_3$, and their relative ratio are adjusted by varying the ratio of total $NO_3^-$ to total $SO_4^{2-}$. The total amount of anions is set to 1 μmol and total ammonia $NH_{3,\ tot}$ is fixed at 2 μmol, ensuring an excess relative to anions. Here we varied the

temperature from 265K to 305K at a step size of 1 K, and the relative humidity from 60% to 95% at a step size of 1%.

**Scenario Met:** This scenario is to investigate the influence of meteorological condition on $\gamma_{AN}^2$ for the $Na^+$-$NH_3$-$H_2SO_4$-$HNO_3$-$H_2O$ system. The total Na is fixed at 5% of the total $SO_4^{2-}$, while the remaining setting is the same as Scenario SNA.

**Scenario Chem:** This scenario is to test the effect of chemical profile on $\gamma_{AN}^2$ over a wider concentration range. The temperature is fixed at 288K, and the relative humidity is fixed at 60%, 75% and 90%. Na varies from 0% to 95% at a step

size of 2%. Remaining variables are the same as Scenario SNA.

**Scenario Full**: This scenario is to compare relative importance of meteorological condition and chemical profile on $\gamma_{AN}^2$ across a comprehensive range of conditions, to fully consider influences of all variables through Sobol's analysis. The temperature range is varied from 265K to 305K at a step size of 5K; the relative humidity range is from 60% to 95% at a step size of 5%. Na accounts for 0~80% of total cations with a step size of 10%. Remaining variables are the same as Scenario SNA.

**2.3 Ambient data**

Long term observational data of inorganic ions ($Na^+$, $SO_4^{2-}$, $NH_4^+$, $NO_3^-$, $Cl^-$, $Ca^{2+}$, $K^+$, $Mg^{2+}$) in $PM_{2.5}$ and gas pollutants ($NH_3$, $HNO_3$, $HCl$) in USA(Edgerton et al., 2006; Hansen et al., 2003) , Canada(Tao and Murphy, 2019) and China(Duan et al., 2024) are collected from published work as detailed in Table S2. For direct comparison, crustal species (e.g., $Ca^{2+}$, $Mg^{2+}$, $K^+$) were transformed into equivalent $Na^+$. In addition, all observational data were harmonized to a uniform temporal resolution, ensuring

that the analysis was consistently conducted on a daily basis.



## 3 Results and Discussions

### 3.1 Influence of $\gamma_{AN}^2$ on nitrate partitioning with different thermodynamic models

The estimated $\gamma_{AN}^2$ across all three thermodynamic models generally fall between $10^{-2}$ and $10^{-1}$. In the $NH_3$-$H_2SO_4$-$HNO_3$-$H_2O$ aerosol system, ISORROPIA constantly predicts $\gamma_{AN}^2$ to be $1.0 \times 10^{-2}$ across all range of chemical compositions and
meteorological conditions (see Fig. 2a). In comparison, $\gamma_{AN}^2$ estimated by the E-AIM (median $\sim 6.1 \times 10^{-2}$) is generally 33% higher than that estimated by AIOMFAC (median $\sim 4.0 \times 10^{-2}$). In the $Na^+$-$NH_3$-$H_2SO_4$-$HNO_3$-$H_2O$ aerosol system, the presence of $Na^+$ shows minor influence on the $\gamma_{AN}^2$ estimation for AIOMFAC and E-AIM. In comparison, after introducing $Na^+$ to system, the $\gamma_{AN}^2$ by ISORROPIA is no longer constant but begins to vary. In general, its $\gamma_{AN}^2$ estimation is slightly ($\sim 8\%$) lower than that of E-AIM, with a median of $\sim 5.6 \times 10^{-2}$ (Fig. 2b).

The differences in $\gamma_{AN}^2$ among the models lead to corresponding variations in $f_{p,NO_3^-}$. Although ISORROPIA align relatively well with E-AIM considering the generally smaller $\gamma_{AN}^2$ differences, the $f_{p,NO_3^-}$ could still differ by $\sim \pm 0.1$. In comparison, AIOMFAC consistently underestimates $\gamma_{AN}^2$ and consequently overestimates $f_{p,NO_3^-}$ as compared with the other two models (Fig. 2c, d). Moreover, the $f_{p,NO_3^-}$ discrepancies depend strongly on the particle-phase preference regime of nitrate, as characterized by the E-AIM predicted $f_{p,NO_3^-}$ here. The estimated $f_{p,NO_3^-}$ differences are generally higher when the E-AIM
predicted $f_{p,NO_3^-}$ values are lower. When the $f_{p,NO_3^-}$ estimated by E-AIM ranged 0.5-0.6, that estimated by AIOMFAC and ISORROPIA could deviate $\sim 0.38$ and $\sim 0.1$, respectively. In comparison, the model discrepancies are nearly negligible at higher $f_{p,NO_3^-}$ values of over 0.9. The large discrepancy between AIOMFAC and the other two models can be largely explained by the absence of gas-phase constraint in its calculations. This may induce large uncertainties, as has been well illustrated in previous studies (Hennigan et al., 2015; Peng et al., 2019; Pye et al., 2020a). In addition to gas-particle partitioning, other
relative variables will also be affected due to different mathematical solutions, see further comparison in SI Text S3 and Figs. S2 - S4.



**Figure 2. Comparisons of $\gamma^2_{AN}$ and $f_{p,NO_3^-}$ among three models,** for (a, c, e) $NH_3$-$H_2SO_4$-$HNO_3$-$H_2O$ system based on Scenario SNA, and (b, d, f) $Na^+$-$NH_3$-$H_2SO_4$-$HNO_3$-$H_2O$ system based on Scenario Met. (a, b) Comparison of estimated $\gamma^2_{AN}$ distributions for different models. (c, d) The $f_{p,NO_3^-}$ estimated by ISORROPIA and AIOMFAC as compared with that estimated by E-AIM. (e, f) Distribution of the mean absolute error (MAE) in estimated $f_{p,NO_3^-}$ with changing E-AIM predicted $f_{p,NO_3^-}$. The boxes and whiskers indicate the 5th, 25th, 50th, 75th and 95th percentiles, respectively.



## 3.2 Influencing factors of $\gamma_{AN}^2$ for Na$^+$-NH$_3$-H$_2$SO$_4$-HNO$_3$-H$_2$O system

As shown in Fig. 2a, ISORROPIA assigns a constant $\gamma_{AN}^2$ of 0.010 for the NH$_3$-H$_2$SO$_4$-HNO$_3$-H$_2$O system. In addition, crustal
ions like Na$^+$ are typically present under ambient conditions. Therefore, below we compared the influencing factors of the
three models with the Na$^+$-NH$_3$-H$_2$SO$_4$-HNO$_3$-H$_2$O system.

In dilute water solution, $\gamma$ is a function of IS only, as described in the Debye-Hückel equation(Zünd, 2007) of:

$$log_{10}\gamma_i = -Az_i^2\sqrt{\text{IS}} \tag{4}$$

where $\gamma_i$ is the activity coefficient of ion $i$, $z_j$ represents charges of ion $i$, and the constant $A$ is a function of temperature and
properties relative to water such as density and static permittivity. The IS ($\mu$mol kg$^{-1}$) is the ionic strength defined as:

$$\text{IS} = \frac{1}{2}\sum m_i z_i^2 \tag{5}$$

Where $m_i$ (mol · kg$^{-1}$ water) is the molality of ion $i$. The IS is an indicator of the overall concentration of ions in solutions,
and is independent of chemical profiles by definition. That is, different ions such as NH$_4^+$ and Na$^+$ would yield the same IS
when they have identical charges and molality.

The IS is mainly determined by $m_i$. In aerosols, the $m_i$ depends largely on AWC, while AWC is modulated mainly by RH and
minorly by chemical species (Seinfeld and Pandis, S. N., 2016; Tan et al., 2017; Zheng et al., 2022). Therefore, we expect a
larger dependence of IS on RH than chemical profiles in aerosols, as also supported in our tests (Fig. S5). Moreover, as the
solutions became highly concentrated, short range forces $F$ (e.g., binary or ternary interactions of ions) begin to play an
important role, which depends on the detailed ionic pairs or the chemical compositions. This would result in the deviation from
the ideal Debye-Hückel equation.

Overall, we show that both meteorological conditions (RH and $T$) and chemical profiles could influence the activity coefficients,
where the RH influence is mainly through the AWC and therefore IS, while the influences of temperature and chemical profiles
are mainly through the thermodynamic equilibrium. The corresponding relationships are illustrated with the interpretive
structural model in Fig. S6. Below we investigated into their detailed influences.

### 3.2.1 Influences of meteorological condition

**Influences of RH and $T$ at given chemical profile.** Figure 3 shows the dependence of $\gamma_{AN}^2$ on temperature and RH based on
the Scenario SNA-Na results. To exclude the influence of particle-phase compositions, here we selected data with $f_{NO3-}$ =0.75
only. As shown in Fig. 3a~c, $\gamma_{AN}^2$ calculated by all three models increases with rising $T$ and RH, while with different
sensitivities. In general, the sensitivity follows the order of E-AIM ≈ ISORROPIA > AIOMFAC, while the sensitivity
difference between AIOMFAC and the other two models is much larger for temperature than for RH. For example, at fixed
RH of 75% while temperature increases from 273 K to 298 K (see black dashed line in Fig. 3), the $\gamma_{AN}^2$ would change by ~0.02
for ISORROPIA and E-AIM, which is 4 times that of AIOMFAC (~0.005). In contrast, for fixed temperature of 298K while
RH increases from 65% to 90% (see white dashed line in Fig. 3), the $\gamma_{AN}^2$ would change by ~0.08 for ISORROPIA and E-AIM,
while that change of AIOMFAC is only slightly smaller (~0.07).



Relative humidity affects $\gamma_{AN}^2$ more strongly than temperature in terms of typical variation ranges under ambient conditions. For example, at a fixed temperature ($T$ = 298 K), varying RH from 65% to 90% ($\Delta$RH = 25%) would lead to an average $\Delta\gamma_{AN}^2$ = 0.075 across all models. However, at a fixed humidity (RH = 75%), increasing temperature from 273 K to 298 K ($\Delta T$ = 25K) only induces an average $\Delta\gamma_{AN}^2$ of 0.015. Our analysis across different timescales further show that RH consistently exerts a stronger influence than $T$ in real atmospheric conditions. In a temperate continental monsoon climate such as Beijing, RH

typically fluctuates by 20 - 40% within a day, while diurnal $T$ variations are around 10 ℃, meaning that humidity changes dominate the daily variability of $\gamma_{AN}^2$. Over seasonal scales, RH differs by about 15 - 25% between summer and winter, whereas $T$ differences can exceed 30 °C; nevertheless, the larger relative impact of RH makes it the primary driver in meteorology of seasonal variability. On even longer timescales (e.g., interannual), annual mean RH varies only within 5 – 10%, while mean $T$ shifts by 1 - 3 °C, again pointing to humidity as the determining factor in meteorology. Therefore, RH dominates the variability

of $\gamma_{AN}^2$ at daily, seasonal, and interannual scales, whereas the role of $T$ is secondary for meteorology.

**Ionic strength as the primary pathway of RH influence on $\gamma_{AN}^2$.** As discussed above, the influence of RH on $\gamma_{AN}^2$ is most likely through IS, which is illustrated in Fig. 3d-f. The general patterns are similar for all the three models. The relationship generally followed the form as outlined in Debye-Hückel law in dilute solutions that $log_{10}\gamma$ is inversely proportional to $\sqrt{IS}$. However, the detailed sensitivity (as quantified by the slope $K$ in $log_{10}\gamma - \sqrt{IS}$ plots; Fig. 3d~f) differs with the particle

compositions $f_{NO3-}$, with higher sensitivity (absolute value of $K$) predicted at higher $f_{NO3-}$ levels. Moreover, the influence of chemical compositions differs much among the three models. E-AIM is the most sensitive model to chemical composition, as reflected in much larger variation of $K$ with $f_{NO3-}$. When $f_{NO3-}$ change from 0.2 to 0.8, the $log_{10}\gamma - \sqrt{IS}$ slope $K$ would change by 0.11 in E-AIM, which is much larger than that in ISORROPIA and AIOMFAC ($K$ changes by ~0.05 and ~0.08, respectively). This indicates a higher sensitivity of $\gamma_{AN}^2$ estimation to chemical profile for the E-AIM model, as also revealed in Sect. 4.3. In

comparison, the $log_{10}\gamma - \sqrt{IS}$ relationship is independent on temperature (see Fig. S7).





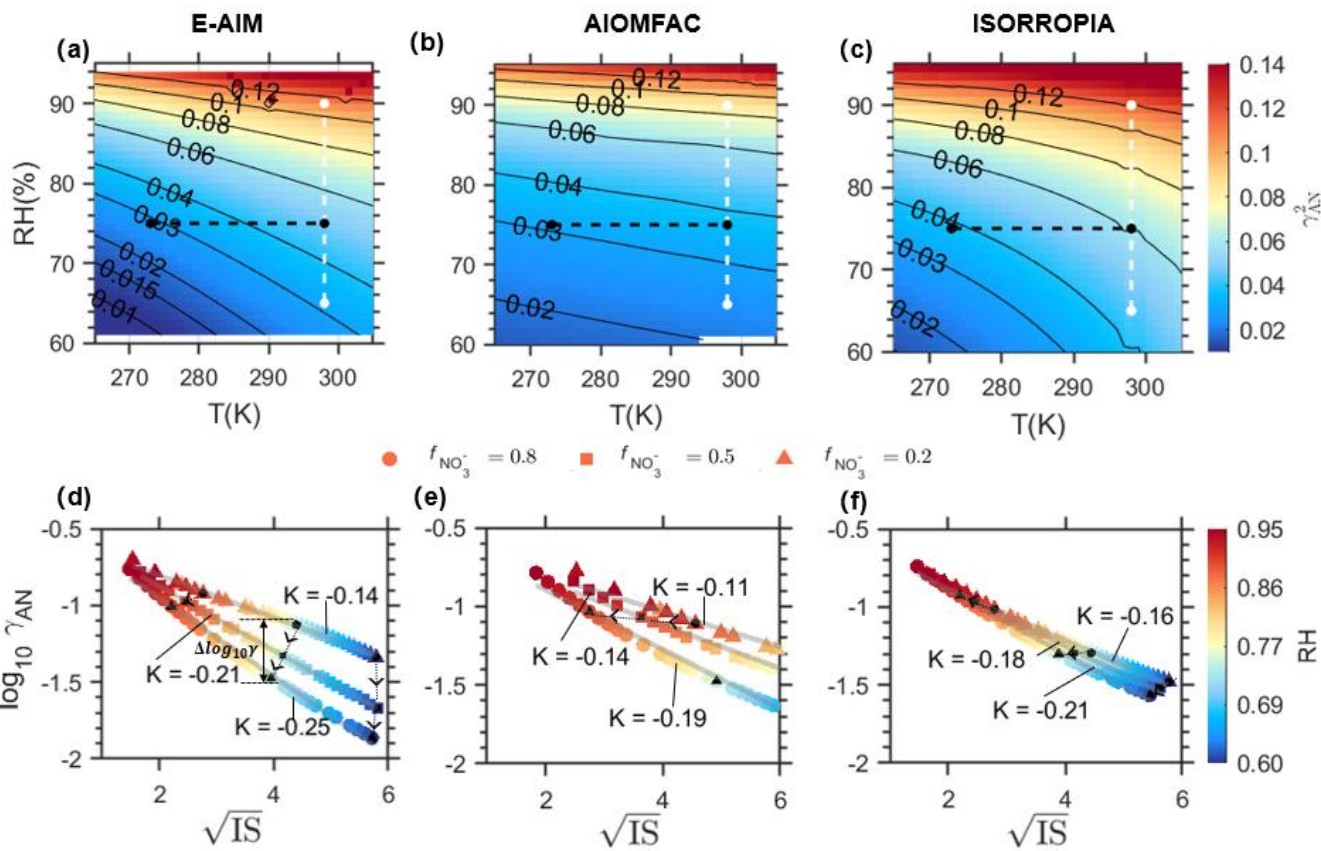

**Figure 3. Comparison of the dependence of $\gamma_{AN}$ on different influencing factors as estimated by (a, d) E-AIM; (b, e) AIOMFAC; (c, f) ISORROPIA.** (a-c) The $\gamma_{AN}^2$ under different T and RH conditions, with $f_{NO3-}$ fixed at 0.75. (d-f) Dependence of $\gamma_{AN}$ to IS and RH at three different $f_{NO3-}$ levels. Here the temperature is fixed at 273 K. Data are based on Scenario Met.


### 3.2.2 Influence of particle-phase chemical profile at given RH and *T* conditions

Figure 4 shows the dependence of $\gamma_{AN}^2$ on particle-phase anion profiles (as characterized by $f_{NO3-}$; sect 2.2) and cation profiles (as characterized by $f_{NVC}$; sect 2.2). Unlike the response to meteorological condition, influence of particle-phase chemical profiles on $\gamma_{AN}^2$ varies markedly among the three thermodynamic models.

The sensitivity of $\gamma_{AN}^2$ to anion profile (or $f_{NO3-}$) is strongly modulated by RH, in terms of both direction and absolute value. The E-AIM predicted a consistently negative correlation of $\gamma_{AN}^2$ - $f_{NO3-}$ across all RH ranges (Fig. 4a). In addition, the magnitude of the correlation weakens substantially from lower to higher RH. For instance, when $f_{NO3-}$ changes from 0.1 to 0.9, the $\Delta\gamma_{AN}^2$ is ~ -0.15 at RH=60%, which weakens to only ~ -0.03 at RH=90%. In contrast, AIOMFAC and ISORROPIA exhibit weak negative correlation at relative lower RH. However, that pattern is reversed to a clear positive correlation at higher RH (e.g., 230   90%) (Fig. 4b, c).





Influence of RH on the sensitivity of $\gamma^2_{AN}$ to cation profile (or $f_{NVC}$) is much weaker (Fig. 4d-f). All three models show positive $\gamma^2_{AN}$ - $f_{NVC}$ correlations at all RH ranges. Yet, the sensitivity shows certain dependence on RH. For E-AIM and AIOMFAC, the sensitivity of $\gamma^2_{AN}$ to $f_{NVC}$ weakens slightly with increasing RH, as indicated by the smaller slopes at higher RH (Fig. 4d, e). In comparison, the $\gamma^2_{AN}$ - $f_{NVC}$ relationship for ISORROPIA remains largely insensitive to RH.

Taken together, the models exhibit much greater divergence in their responses to anion perturbations than to cation perturbations, highlighting substantial uncertainties in thermodynamic predictions of $\gamma^2_{AN}$ under varying aerosol particle phase chemical profile. Notably, E-AIM shows the highest sensitivity to chemical profiles, in terms of both anions and cations (see Table S3).

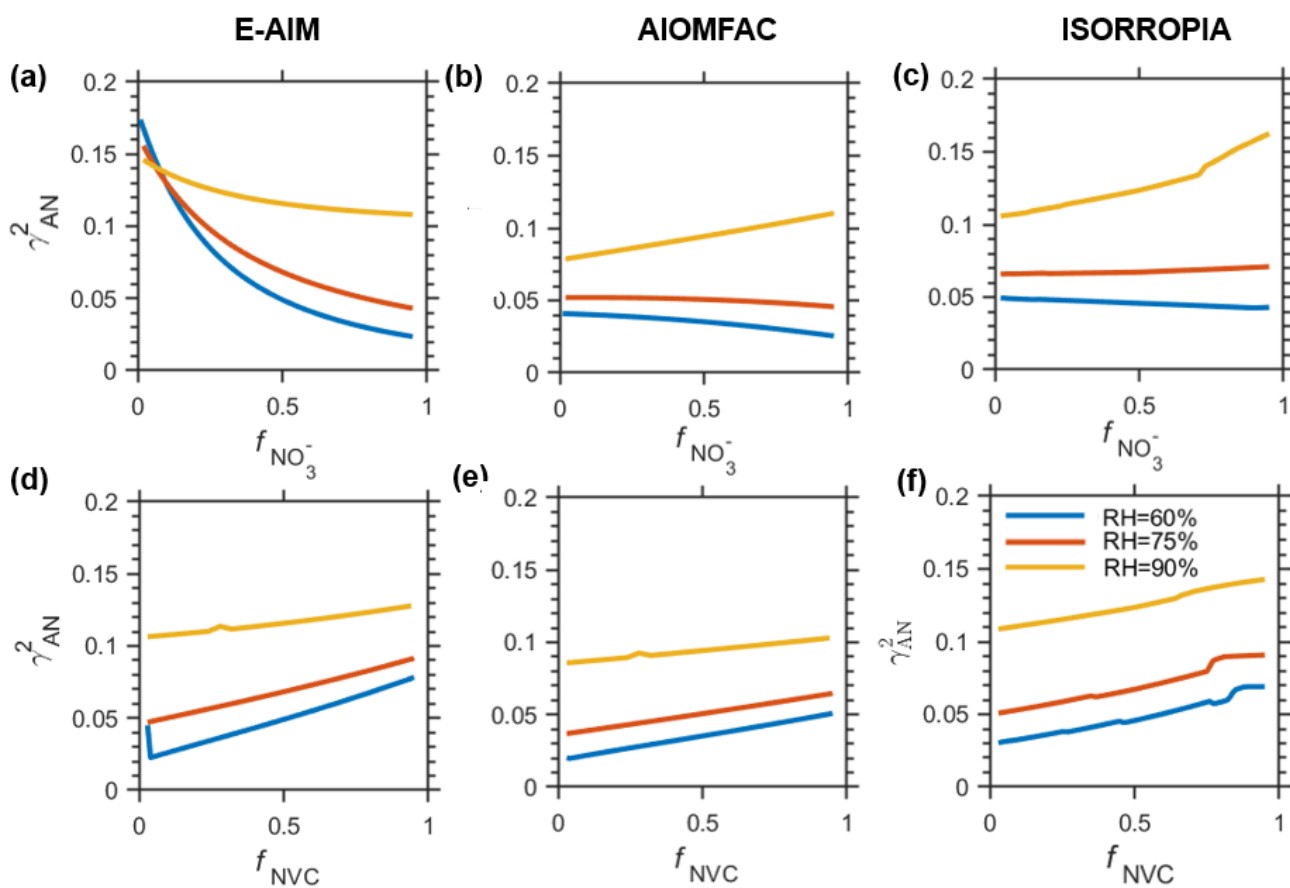

**Figure 4. The $\gamma^2_{AN}$ under different (a~c) $f_{NO3-}$ and (d~f) $f_{NVC}$** estimated by (a), (d) E-AIM; (b), (e) AIOMFAC; (c), (f) ISORROPIA when the opposite ions ($f_{NVC}$ / $f_{NO3-}$) is fixed at 0.5. RH=90%, 75%, 60% are selected to represent different RH levels. Data are based on Scenario Chem.



### 3.2.3 Relative importance of meteorological condition vs. chemical profile

To examine the overall relative importance of meteorological condition and chemical profile on $\gamma_{AN}^2$, we adopted the Sobol's variance decomposition method(Feinberg et al., 2020; Ji et al., 2018). This method is a global sensitivity analysis approach that partitions the variance of a model output into contributions from individual input factors and their interactions, thereby quantifying how much each factor and their combinations influence the output variability and thus determining their relative importance within a given model. Note that Sobol's variance decomposition method requires all input variables must be

statistically independent of each other (see Fig. S8). Therefore, we selected the key parameters of RH and temperature, $f_{NO3-}$ and $f_{NVC}$ to represent the meteorological conditions and chemical profiles, respectively.

The results show that for AIOMFAC and ISORROPIA, $\gamma_{AN}^2$ variations are largely regulated by RH rather than chemical profile (see Table 1). In comparison, for E-AIM the $\gamma_{AN}^2$ is more sensitive to chemical profiles than meteorology. Especially, E-AIM show the largest sensitivity to the anion profiles $f_{NO3-}$, which is consistent with the results presented in Fig. 4a, d.

We also note that while E-AIM is less sensitive to metrological conditions than to chemical profile, its absolute sensitivity to meteorological factor is still comparable to ISORROPIA and substantially higher than that of AIOMFAC, especially in terms of temperature (sect. 4.1; Fig. 3a~c)(Pye et al., 2020b). As E-AIM is typically treated as the benchmark model, these results implies that the ISORROPIA could roughly capture the influence of meteorological conditions on $\gamma_{AN}^2$, while its representation on the influence of chemical profiles is not enough. In comparison, the AIOMFAC needs to be improved in the considerations

of both meteorological conditions and chemical profiles.

**Table 1. Sobol's variance decomposition of different factors based on Scenario Full.**

| Model | Total Sobol' indices[*] | | | |
|---|---|---|---|---|
| | RH | T | $f_{NO3-}$ | $f_{NVC}$ |
| E-AIM | 0.42 | 0.16 | 0.54 | 0.22 |
| AIOMFAC | 0.97 | 0.14 | 0.03 | 0.01 |
| ISORROPIA | 0.72 | 0.09 | 0.04 | 0.18 |

[*] Total Sobol' indices are used in global sensitivity analysis to quantify the contribution of an input variable and its interaction effects with other variables to the total variance of the model output.

### 3.3 Dominant influencing factors for ambient aerosols

The dependences of $\gamma_{AN}^2$ to different influencing factors as estimated by the three thermodynamic models are further evaluated with ambient observations worldwide. Overall, the $\gamma_{AN}^2$ range from 0.008 to 0.3 (see Fig. 5). The $\gamma_{AN}^2$ as predicted with E-AIM are generally higher than the other two models, in agreement with the results from the simulation data (Fig. 2a, b). Consequently, E-AIM estimates a lower $f_pNO_3^-$ than the other two models (see Fig. 5d). AIOMFAC occasionally yields $f_pNO_3^-$

values outside the physically valid range of 0~1(<2%), indicating that further improvements are needed in the current version





of AIOMFAC for reliable gas–particle partitioning predictions. However, none of them are in good alignment with observational $f_p\text{NO}_3^-$, and larger underestimation is often seen in lower ambient $f_p\text{NO}_3^-$ range (see Fig.S9).

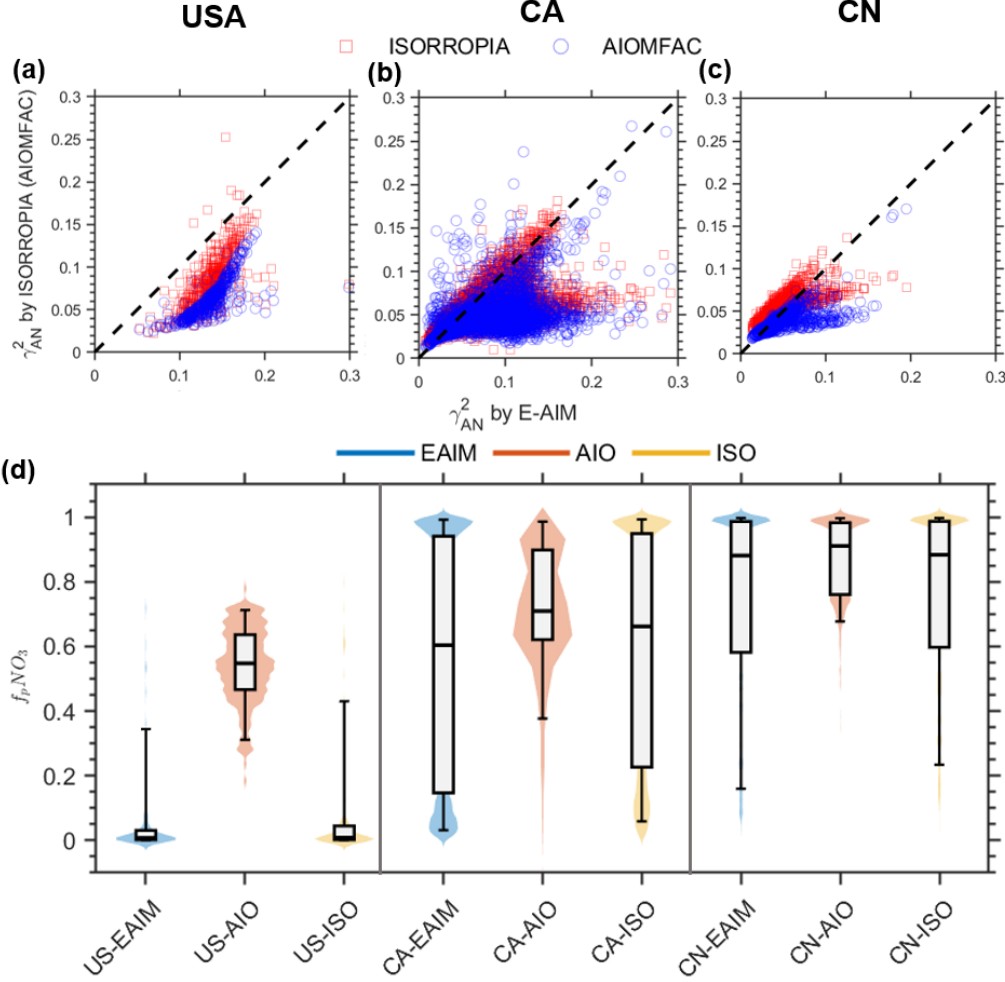

**Figure 5. Estimation of (a~c) $\gamma^2_{AN}$, (d) $f_p\text{NO}_3^-$ from three thermodynamic models, based on global observational data.** The (a) left, (b) middle, and (c) right panels correspond to the USA, Canada (CA), and China (CN), respectively. Violin-box plots of $\gamma^2_{AN}$ simulated by three thermodynamic models (EAIM, AIOMFAC, ISORROPIA) under three regions (USA, CA, CN). The shaded violin background indicates the probability density of the data distribution. The boxes and whiskers indicate the 5th, 25th, 50th, 75th and 95th percentiles, respectively.

Sobol's variance decomposition analysis corroborates the simulation findings, indicating that chemical profiles are the primary controlling factor in E-AIM, whereas meteorological conditions play a more significant role in ISORROPIA and AIOMFAC (see Table 2). Furthermore, the relative influence of anions versus cations varies with location. As can be seen from E-AIM



results, while anion profiles exert a stronger effect in the USA, cation profiles are more dominant in Canada and China. These results reveal that the controlling factors for $\gamma^2_{AN}$ are model-dependent and location-specific.


**Table 2 Sobol's variance decomposition of different factors based on observational data**

| Country | Model | Total Sobol' indices[*] | | | | Dominating factor |
|---|---|---|---|---|---|---|
| | | RH | $T$ | $f_{NO3-}$ | $f_{NVC}$ | |
| USA | E-AIM | 0.16 | 0 | 0.61 | 0.44 | Chemical |
| | AIOMFAC | 0.87 | 0.05 | 0.01 | 0.09 | Meteorological |
| | ISORROPIA | 0.65 | 0.35 | 0.02 | 0.03 | Meteorological |
| Canada | E-AIM | 0.11 | 0 | 0.11 | 0.87 | Chemical |
| | AIOMFAC | 0.86 | 0.07 | 0.03 | 0.12 | Meteorological |
| | ISORROPIA | 0.07 | 0.8 | 0 | 0.4 | Meteorological |
| China | E-AIM | 0.42 | 0.16 | 0.36 | 0.52 | Chemical |
| | AIOMFAC | 0.53 | 0.46 | 0 | 0 | Meteorological |
| | ISORROPIA | 0.75 | 0.04 | 0.02 | 0.22 | Meteorological |

## 4 Conclusions

Our results show significant differences of $\gamma^2_{AN}$ and $f_p NO_3^-$ estimation among three widely-used thermodynamic models, i.e.
ISORROPIA, E-AIM, and AIOMFAC. While the E-AIM is typically considered as the benchmark, ISORROPIA is more widely adopted in 3-D chemical-transport models, whereas AIOMFAC is preferred in dealing with organic-related processes. The large difference among these models indicate that model choice can substantially influence the predicted particle-phase activity coefficient and nitrate partitioning, which may bring non-negligible uncertainties and can be important in explaining the gaps among observations, chamber studies and large-scale model simulations.

While all three models show strong dependence of $\gamma^2_{AN}$ on RH, their estimation of the $\gamma^2_{AN}$ dependence on chemical profiles differed much. Especially, while for E-AIM the $\gamma^2_{AN}$ is more sensitive to chemical profiles, for ISORROPIA and AIOMFAC the meteorological conditions play the major role. These results indicate the needs for improved consideration of chemical profiles in $\gamma^2_{AN}$ estimations, especially for ISORROPIA and AIOMFAC. More chamber and ambient observations, as well as theoretical calculations are encouraged in future studies to derive a unified and comprehensive picture, and therefore to
improve the accuracy of aerosol thermodynamic predictions and better inform air quality and climate assessments.

**Author contribution**

**Wan Ruilin**: Software, Investigation, Writing- Original draft preparation. **Zheng Guangjie**: Conceptualization, Methodology, Writing- Original draft preparation, Writing - Review & Editing, Project administration. **Li Yuyang and Duan Xiaolin:**



Validation, Investigation, Writing - Review & Editing. **Jiang Jingkun and He Kebin**: Supervision, Project administration, Writing - Review & Editing

## Competing interests

At least one of the co-authors is member of the editorial board of Atmospheric Chemistry and Physics.

## Acknowledgements

The research was supported by the National Natural Science Foundation of China (22476106 and 22188102).

## Data availability

Data are available on request.

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
