# Peer review of "Driving factors for the activity coefficient of atmospheric ammonium nitrate: discrepancies among thermodynamic models and impact on nitrate pollutions"

_EGUsphere, 2025_

## Author Comment (AC1)

**Comments by Anonymous Referee #1**

*General Comments*:

*This study investigates the driving factors of ammonium nitrate activity using comprehensive simulations and global ambient observations, comparing three widely used thermodynamic models (ISORROPIA, E-AIM, and AIOMFAC) to clarify the impacts of meteorological conditions and chemical profiles.*

*The manuscript is acceptable for publication following the implementation of the following key revisions:*

**Responses:**

We thank the reviewer for the comments. Please find our point-to-point responses below.

*Specific comments:*

1. *How does this study interpret the activity coefficient of ammonium nitrate (NH₄NO₃)? Aerosols are complex systems, and the individual activity coefficients of NH₄⁺ and NO₃⁻ are objectively existing concepts. How is the activity coefficient of NH₄NO₃ defined to clarify its differences from those of other compounds such as sodium nitrate (NaNO₃) and ammonium sulfate ((NH₄)₂SO₄)? Additionally, why does the study use γₐₙ in some places and its square in others?*

**Responses:**

We thank the reviewer for the comment. The mean activity coefficients of neutral electrolytes is typically defined as (Zünd, 2007):

$$\gamma_{\pm} = \left[ \gamma_+^{v^+} \cdot \gamma_-^{v^-} \right]^{1/(v^+ + v^-)} \qquad \text{(R1)}$$

This concept is introduced as an important complement of the activity coefficients of individual cations and anions ($\gamma_+$ or $\gamma_-$), as only $\gamma_{\pm}$ can be directly measured. This is due to that any real solution must be electrically balanced. The $\gamma_+$ or $\gamma_-$ cannot be measured, but is derived or calculated based on the measured $\gamma_{\pm}$ values under different conditions.

Following this convention, here the "activity coefficient of ammonium nitrate" ($\gamma_{AN}$) represents the mean ionic activity coefficient of the dissociated ion pair NH₄⁺ and NO₃⁻, which is defined as:

$$\gamma_{AN} = \sqrt{\gamma_{NH_4^+} \gamma_{NO_3^-}}. \qquad \text{(R2a)}$$

Or equivalently,

$$\gamma_{AN}^2 = \gamma_{NH_4^+} \gamma_{NO_3^-} \qquad \text{(R2b)}$$

Similarly, the activity coefficient of sodium nitrate (NaNO₃) and ammonium sulfate ((NH₄)₂SO₄) would be defined as $\gamma_{SN} = \sqrt{\gamma_{Na^+} \gamma_{NO_3^-}}$ and $\gamma_{AS} = [\gamma_{NH_4^+} \cdot \gamma_{SO_4^{2-}}^2]^{1/3}$ , respectively. In a mixed solution of NaNO₃ and NH₄NO₃, for example, the $\gamma_{NO_3^-}$ is the same, and their difference would be caused by the cation, or $\gamma_{AN}^2 / \gamma_{SN}^2 = \gamma_{NH_4^+} / \gamma_{Na^+}$.

For the three thermodynamic models of concern, ISORROPIA can only output the mean activity coefficient $\gamma_{\pm}$, while E-AIM and AIOMFAC can estimate individual cation and anion activity coefficient. As shown in Eq. R2b, the $\gamma_{AN}^2$ is adopted for consistency and easy comparison among the three models. In comparison, the $\gamma_{AN}$ is used only for definition, or when referring to "the activity coefficient of ammonium nitrate".

We've further clarified this point in the revised manuscript as (see the last paragraph in Introduction):

"Our previous studies have revealed that the mean activity coefficient of ammonium nitrate, $\gamma_{AN} = \sqrt{\gamma_{NH_4^+}\gamma_{NO_3^-}}$, is a key parameter influencing the gas-particle partitioning of nitrate (see SI Text S1) (Zheng et al., 2022). Note that for easy comparison with individual ions and among different thermodynamic models, the square form of $\gamma_{AN}$, or $\gamma^2{}_{AN} = \gamma_{NH_4^+}\gamma_{NO_3^-}$, is adopted in following discussions (Zheng et al., 2022)."

2. *I understand that the authors did not decouple meteorological factors and chemical composition in the study design. However, the expression in lines 14–16 of the abstract may need further refinement to more clearly convey the interdependence and relative contributions of these two types of factors, thus avoiding potential ambiguity for readers regarding the study's core findings on $\gamma_{AN}$.*

**Responses:**

We thank the reviewer for the comment. We've further refined the statement into:

"For all three models and all chemicals profile tested, the $\gamma_{AN}^2$ correlates positively with relative humidity (RH) and temperature, and RH generally contributes larger variations . In comparison, the effect of chemical composition on $\gamma_{AN}^2$ is more complex and is strongly modulated by RH, with differed dependence pattern observed at varying RH levels."

3. *The title mentions the "impact on nitrate pollutions", yet the relevant description in the abstract is overly brief. It is recommended to supplement a concise statement explaining how discrepancies in ammonium nitrate activity coefficients among thermodynamic models affect the prediction, assessment, or mitigation of nitrate pollution. This will help readers quickly grasp the real-world relevance of the research beyond theoretical parameter analysis.*

**Responses:**

We thank the reviewer for the comment. We've added relevant statement in the revised abstract as:

"The activity coefficient of $NH_4NO_3$, $\gamma_{AN}$, is one key parameter controlling the gas-particle partitioning of nitrate, with lower $\gamma_{AN}$ typically favoring particle-phase partitioning of nitrate. However, the $\gamma_{AN}$ dependence on meteorological condition and chemical profile remains uncertain."

In addition, we've clarified this point in the manuscript as (see Line 71-76 in the revised manuscript):

"Our previous studies have revealed that the mean activity coefficient of ammonium nitrate, $\gamma_{AN} = \sqrt{\gamma_{NH_4^+}\gamma_{NO_3^-}}$, is a key parameter influencing the gas-particle partitioning of nitrate, with lower $\gamma_{AN}$ typically favoring higher particle-phase partitioning of nitrate (see SI Text S1)(Zheng et al., 2022). This can be interpreted in that, the lower activity coefficient would reduce the activity of nitrate at given concentrations, while it's the activity that matters in the gas-particle equilibrium. Therefore, at given gas-phase concentrations, the equilibrium activity is fixed, while the actual particle-phase concentration would increase with decreased activity coefficient $\gamma$. Note that for easy comparison with individual ions and among different thermodynamic models, the square form of $\gamma_{AN}$, or $\gamma^2{}_{AN} = \gamma_{NH_4^+}\gamma_{NO_3^-}$, is adopted in following discussions (Zheng et al., 2022)."

4. *It is recommended that 2–3 sentences be added in the Introduction to summarize the comparisons of the three thermodynamic models (ISORROPIA, E-AIM, and AIOMFAC) regarding pH and hydrogen ion activity. This supplementation will help better highlight the research gap in the comparative analysis of $\gamma_{AN}$ and clarify the necessity of the current study.*

**Responses:**

We thank the reviewer for the comment. Below has been added into Introduction line 79-83.

"Previous studies on thermodynamic model comparison and performance evaluations on non-ideality characterizations focused primarily on acidity (i.e., the activity coefficient of $H^+$) (Liu et al., 2017; Peng et al., 2019; Song et al., 2018; Yao et al., 2006; Zheng et al., 2022). These studies have shown that ISORROPIA, E-AIM, and AIOMFAC can yield systematically different predictions of aerosol pH under identical chemical and meteorological conditions, partially due to differences in their estimation of ion activity coefficients including $\gamma_{H+}$ and $\gamma_{AN}^2$. Despite these documented discrepancies in acidity-related diagnostics, a comparable inter-model evaluation of the ammonium nitrate activity coefficient and its sensitivity to chemical and meteorological drivers remains scarce."

5. *The statement "AIOMFAC consistently underestimates..." in line 147 is overly absolute.*

**Responses:**

We thank the reviewer for the comment. We've revised accordingly as (see Line 163 in revised manuscript):

"Although ISORROPIA align relatively well with E-AIM considering the generally smaller $\gamma_{AN}^2$ differences, the $f_{p,NO_3^-}$ could still differ by $\sim \pm 0.1$. In comparison, AIOMFAC tends to underestimates $\gamma_{AN}^2$ and consequently overestimates $f_{p,NO_3^-}$ as compared with the other two models."

---

## Author Comment (AC2)

**Comments by Anonymous Referee #2**

*General Comments*:

*Ammonium nitrate (AN) is an important inorganic aerosol, impacting air quality and climate. However, the activity coefficient of AN that shapes the gas-particle partitioning has not been well examined. This paper investigated the dependence of the activity coefficient of AN on meteorological conditions and chemical composition by using three commonly used thermodynamic models. The critical role of RH or ionic strength was demonstrated through well-designed aerosol proxies and real ambient aerosols. The findings have important implications for improving the prediction of ammonium nitrate aerosols. Overall, the paper is well written and fits the scope of ACP. The paper can be recommended for publication after addressing the following comments.*

**Responses:**

We thank the reviewer for the comments. Please find our point-to-point responses below.

*Specific comments:*

1. *Abstract: I would suggest the authors add 1-2 sentences to conclude the impact of uncertainties of the activity coefficient of AN on nitrate pollutions, given that the impact on nitrate pollutions has been highlighted in the title.*

**Responses:**

We thank the reviewer for the comment. We've added relevant statement in the revised abstract as: "The activity coefficient of $NH_4NO_3$, $\gamma_{AN}$, is one key parameter controlling the gas-particle partitioning of nitrate, with lower $\gamma_{AN}$ typically favoring particle-phase partitioning of nitrate. However, the $\gamma_{AN}$ dependence on meteorological condition and chemical profile remains uncertain."

In addition, we've clarified this point in the manuscript as (see Line 70-77 in the revised manuscript): "Our previous studies have revealed that the mean activity coefficient of ammonium nitrate, $\gamma_{AN} = \sqrt{\gamma_{NH_4^+}\gamma_{NO_3^-}}$, is a key parameter influencing the gas-particle partitioning of nitrate, with lower $\gamma_{AN}$ typically favors higher particle-phase partitioning of nitrate (see SI Text S1)(Zheng et al., 2022). This can be interpreted in that, the lower activity coefficient would reduce the activity of nitrate at given concentrations, while it's the activity that matters in the gas-particle equilibrium. Therefore, at given gas-phase concentrations, the equilibrium activity is fixed, while the actual particle-phase concentration would increase with decreased activity coefficient $\gamma$. Note that for easy comparison with individual ions and among different thermodynamic models, the square form of $\gamma_{AN}$, or $\gamma^2_{AN} = \gamma_{NH_4^+}\gamma_{NO_3^-}$, is adopted in following discussions (Zheng et al., 2022)."

2. *Lines 50-60: It is unclear how the different methods of calculating activity coefficients for three models would introduce intrinsic differences in activity coefficients. More discussions on this aspect would help understand the different performances of the models.*

**Responses:**

We thank the reviewer for the comment. To address this concern, we have revised line 53-61 as below:

"The ISORROPIA employs an extended Debye-Hückel form ("Bromley's formula"), in which non-ideality is parameterized through empirical ion-pair terms. While computationally efficient, this approach assumes simplified binary ion interactions and is known to become less accurate at elevated ionic strengths of above ~6 mol kg$^{-1}$(Bromley, 1973; Nenes et al., 1998). The E-AIM calculated $\gamma$ for individual ions based on the Pitzer–Simonson–Clegg formula, which accounted for long-range electrostatic interactions via Debye-Hückel effect and short-range binary/ternary ion–ion interactions through a Margules expansion(Clegg et al., 1992; Pitzer and Simonson, 1986), with parameters from empirical data (Carslaw et al., 1995; Clegg et al., 1998b; Friese and Ebel, 2010). This structure enables E-AIM to better capture non-ideal behavior in highly concentrated electrolyte solutions. AIOMFAC combines a Pitzer-like electrolyte model with a modified UNIFAC approach, representing long-, middle-, and short-range organic–inorganic interactions, allowing for explicit treatment of more organic–inorganic interactions (Zuend et al., 2010; Zünd, 2007). E-AIM and ISORROPIA …..."

3. *Lines 118-119: The amount of species in μg/m3 can be provided to have a straightforward connection with ambient conditions.*

**Responses:**

We thank the reviewer for the comment. Now it has been revised as below in line 132-134:

"The total amount of anions is set to 1 μmol m$^{-3}$, corresponding to approximately 62–96 μg m$^{-3}$ depending on anion composition (e.g., NO$_3^-$ versus SO$_4^{2-}$), NH$_{3,\ tot}$ is fixed at 2 μmol m$^{-3}$ (34 μg m$^{-3}$), ensuring an excess relative to anions."

4. *Line 155: As shown in Fig. S3, chloride is present in ISORROPIA, but not in EAIM. Why? How would chloride influence the calculations of activity coefficients?*

**Responses:**

We thank the reviewer for the comment. The presence of trace chloride in ISORROPIA originates from the internal structure of the model rather than from the input aerosol composition. For the Na$^+$-NH$_3$-H$_2$SO$_4$-HNO$_3$-H$_2$O system, ISORROPIA automatically invokes the ISRP3F subroutine, which is formulated for sodium–ammonium–nitrate–sulfate–chloride aerosol systems (Fountoukis and Nenes, 2007; Nenes et al., 1998). In this subroutine, Cl$^-$ is treated as a potential aqueous species and the equilibrium equations are solved using a bisection method by scanning a range of very low [Cl$^-$(aq)] values to ensure numerical robustness. As a result, trace amounts of chloride may appear in the output even when no chloride is specified in the input.

The predicted chloride concentrations (10$^{-5}$ μmol m$^{-3}$) are several orders of magnitude smaller than the dominant anions NO$_3^-$ and SO$_4^{2-}$ (10$^{-1}$ μmol m$^{-3}$) (see Fig. S3), contributing negligibly to ionic strength. Consequently, their influence on the calculated activity coefficients is insignificant and does not affect the conclusions of this study. In contrast, E-AIM strictly solves the equilibrium based on the specified input species, and chloride is absent when it is not in the input.

5. *Lines 186-187: Is the dependence on RH and T similar for other conditions with different fractions of nitrate?*

**Responses:**

We thank the reviewer for the comment. To examine the robustness of the RH and $T$ dependence under different nitrate fractions, we have added additional cases with $f_{NO3-}$ = 0.6 and 0.9, which are presented in Fig. S7 of the Supplement. The results show that the overall dependence on RH and temperature remains qualitatively similar across the different nitrate fractions.

We've also clarified this point in the abstract as:

"For all three models and all chemicals profile tested, the $\gamma_{AN}^2$ correlates positively with relative humidity (RH) and temperature, and RH generally contributes larger variations ."

6. *Section 3.2.2: Have the authors examined the dependence of $\gamma_{AN}^2$ on sulfate concentrations? The presence of other soluble ions can largely mediate the calculation of activity coefficients.*

**Responses:**

We thank the reviewer for the comment. The dependence of $\gamma_{AN}^2$ on sulfate is implicitly examined in the chemical profile tests. In our setup, the total amount of anions (i.e., $[NO_3^-(p)+2[SO_4^{2-}(p)])$ is fixed at 1 $\mu mol\ m^{-3}$, and the nitrate fraction within the anion pool is defined as:

$$f_{NO_3^-}\left(\frac{\mu eq}{\mu eq}\right) = \frac{[NO_3^-(p)]}{[Anions(p)]} = \frac{[NO_3^-(p)]}{[NO_3^-(p)] + 2[SO_4^{2-}(p)]}$$

Therefore, when $f_{NO_3^-}$ is varied, the sulfate fraction changes accordingly. As a result, the influence of sulfate on $\gamma_{AN}^2$ is inherently embedded in the analysis through the varying anion composition.

7. *Lines 371-372: The underestimation may also arise from the uncertainties of measured $f_{pNO3}$. The authors should provide more discussion on this aspect and offer additional insights on how to narrow the discrepancies between measurements and models.*

**Responses:**

We thank the reviewer for the comment. We've revised the manuscript accordingly as (see line 286-298 in the revised manuscript):

"However, none of them are in good alignment with observational $f_pNO_3^-$, and larger underestimation is often seen in lower ambient $f_pNO_3^-$ range (see Fig. S10). This may also be partially attributed to the uncertainties of measured $f_{pNO3}$, including sampling artifacts associated with semi-volatile ammonium nitrate, potential volatilization losses during filter-based measurements, temporal mismatches between gas-phase $HNO_3$ and particulate $NO_3^-$ observations, etc. These effects can be particularly pronounced under low total nitrate ($NO_3^-$ + $HNO_3$) conditions, where small absolute errors in nitrate or nitric acid measurements may translate into large uncertainties in $f_pNO_3^-$ (Guo et al., 2016; Tao and Murphy, 2019). Future studies should therefore focus on narrowing these discrepancies through coordinated improvements in both measurement and model. On the measurement side, the use of online or semi-continuous techniques, together

with collocated and time-resolved observations of gas-phase $HNO_3$ and particulate $NO_3^-$, would help reduce uncertainties associated with sampling artifacts and temporal mismatches. On the modeling side, the variability of $f_pNO_3^-$, especially at low nitrate levels, may be better captured by considering potential kinetic limitations and by improving the parameterization of activity coefficients in inorganic-organic mixed aerosol system. Observation-constrained modeling, together with sensitivity analyses, can further reduce discrepancies in $f_pNO_3^-$ between modeled and observed values."